# ADHD Prevalence among Outpatients with Severe Opioid Use Disorder on Daily Intravenous Diamorphine and/or Oral Opioid Maintenance Treatment

**DOI:** 10.3390/ijerph20032534

**Published:** 2023-01-31

**Authors:** Henrik Rohner, Nikolas Gaspar, Helena Rosen, Tim Ebert, Laura Luisa Kilarski, Felix Schrader, Moaz Al Istwani, Anna Julia Lenz, Christoph Dilg, Andrea Welskop, Tatjana Goldmann, Ulrike Schmidt, Alexandra Philipsen

**Affiliations:** 1Department of Psychiatry and Psychotherapy, Faculty of Medicine, University of Bonn, 53111 Bonn, Germany; 2Department of Psychiatry and Psychotherapy, Faculty of Medicine, University Medical Center of Goettingen, 37075 Goettingen, Germany

**Keywords:** ADHD, diamorphine, SUD, prevalence

## Abstract

(1) Background: Attention deficit hyperactivity disorder (ADHD) is a common comorbid condition in opioid use disorder (OUD) and is associated with a more severe course of substance use. Patients with severe OUD who have not responded to oral opioid maintenance treatment can be treated with intravenous diamorphine up to three times per day. Here, we investigated the prevalence of ADHD among patients undergoing either daily diamorphine maintenance treatment or daily oral opioid maintenance treatment. (2) Methods: We assessed all participants with the WURS-k and the ADHD-SR. The Diagnostic Interview for ADHD in Adults (DIVA) was performed with all participants who met the cut-off in the WURS-k and/or ADHD-SR. (3) Results: The overall prevalence of ADHD was 17.9%. Prevalence of ADHD among patients undergoing daily diamorphine maintenance treatment was 14.3%. Prevalence of ADHD among patients undergoing daily oral opioid maintenance treatment was 20.3%. The combined presentation of ADHD was the most prevalent condition. In urine samples of participants with comorbid ADHD, heroin was detected the most and cocaine the least frequently. (4) Conclusions: Almost one out of five patients with OUD suffered from comorbid ADHD. In 83.3%, ADHD had not been diagnosed prior to participation in this study. Thus, patients with SUD could benefit from being routinely screened for ADHD.

## 1. Introduction

Attention deficit hyperactivity disorder (ADHD) has a prevalence of 2.5% in adults and 3.4% in childhood and is one of the most prevalent neurodevelopmental disorders among children and adolescents [1,2]. ADHD is characterized by a persistent pattern of inattention and/or hyperactivity/impulsivity that results in functional impairment in multiple settings, which leads to behavior comparably more inappropriate or disruptive than in other people of similar age [3]. Usually, ADHD symptoms become apparent in childhood; however, many of the affected patients continue to experience ADHD symptoms in adolescence and adulthood [4]. According to the Diagnostic and Statistical Manual of Mental Disorders 5 (DSM-5), there are three predominant presentations of ADHD [5]. In adult ADHD populations, the combined presentation is the most prevalent (70%), followed by the predominantly inattentive presentation (18.3%) and the predominantly hyperactive-impulsive presentation (8.3%) [6].

ADHD has been associated with a more severe course of substance use as well as with social and mental health impairment [7]. Moreover, patients with ADHD are more likely to develop a substance use disorder (SUD) [8] at a younger age [9].

An expert group suggests starting the diagnostic process for ADHD among SUD patients as soon as possible, when there are neither serious withdrawal symptoms nor serious intoxication [10].

The prevalence of ADHD in SUD populations has been found to be inconsistent ranging from 5.22% [11] to 62% [12]. A meta-analysis reported a prevalence of ADHD of 23.1% in alcohol and opioid use disorder, both showing a higher prevalence than in cocaine use disorder [13]. A recent study reported a prevalence of 18.2% for ADHD persisting into adulthood in populations of patients with opioid use disorder (OUD) [14]. Furthermore, a recent meta-analysis reported a ADHD prevalence of 20.9% in populations of OUD patients [15].

OUD patients with comorbid ADHD that undergo oral opioid maintenance treatment are characterized by greater addiction severity and higher psychiatric comorbidity rates [16]. Most of these patients are addicted to diacetylmorphine (diamorphine), better known as heroin. Since 2010, OUD patients can legally switch from oral opioid maintenance treatment to intravenous diamorphine maintenance treatment in specialized outpatient clinics if they fulfil certain criteria. These include being at least 23 years of age, having used heroin intravenously for at least five years in total and suffering from comorbid mental or physical illnesses [15]. Switching to diamorphine maintenance treatment has been shown to improve physical and mental health of this group of patients as well as their social wellbeing [17]. To the best of our knowledge, the prevalence of ADHD in diamorphine patients has not yet been evaluated.

Hence, the primary objective of this study was to contribute to closing this knowledge gap by determining, for the first time, the prevalence of ADHD in a population of adult patients with severe opioid use disorder (OUD) undergoing daily intravenous diamorphine maintenance treatment in Germany. In addition, for comparison, we examined a population of adult patients with OUD undergoing daily oral opioid maintenance treatment in the same city.

## 2. Materials and Methods

### 2.1. Procedures and Study Design

We investigated two populations of patients with OUD recruited in two outpatient departments in Bonn, Germany; namely, a population treated at the *Diamorphinambulanz* of the university hospital of Bonn with daily intravenous diamorphine maintenance treatment (DIA) and another population treated at the *Medizinische Ambulanz und Substitutionsbehandlung* of the *Verein für Gefährdetenhilfe* (VfG) treated with daily oral opioid maintenance treatment (OOT).

Patient recruitment started after ethical approval of the local Ethics Committee of the University Clinic of Bonn had been obtained.

Every patient of these outpatient departments was addressed and informed about the aims and procedure of this study. Inclusion criteria were minimum age of 18 years and a diagnosis of opioid dependence according to the International Statistical Classification of Diseases and Related Health Problems–10 (ICD-10) [18]. Exclusion criteria were severe withdrawal and/or intoxication symptoms. Furthermore, somatic or psychiatric conditions leading to serious cognitive deficits, such as Korsakoff syndrome or acute psychosis, also led to exclusion.

Diagnosis of ADHD followed the guidelines of the National Institute for Health and Care Excellence (NICE) [19]. After written confirmed consent had been obtained, the German version of the Wender Utah Rating Scale (WURS-k) [20] and the German self-rating behavior questionnaire (ADHD-SR) [21] were completed by all participants. The Diagnostic Interview for ADHD in Adults (DIVA 2.0) [22] was conducted with all participants that met the cut-off criteria in the WURS-k and/or the ADHD-SR. Furthermore, we checked the latest urine sample of every participant with a positive result in the DIVA 2.0 Interview for heroin, benzodiazepines, cocaine, amphetamine, and cannabis.

Urine samples are usually analyzed monthly for every patient undergoing opioid maintenance treatment. The urine samples of DIA patients are analyzed for illicit and pharmaceutical diamorphine/heroin use by gas chromatography-mass spectrometry [23].

### 2.2. Psychometric Inventories

The WURS-k is a retrospective questionnaire about ADHD symptoms between the ages of 8 and 12 years. ADHD-SR is a questionnaire about ADHD symptoms in adulthood. Cut-off criteria in WURS-k is 30 out of 84 points and cut-off criteria in ADHD-SR is 18 out of 66 points. Combined, these two questionnaires have a sensitivity of 94% and specificity of 56% [24].

DIVA 2.0 is a clinical diagnostic interview for ADHD in childhood and adulthood in line with the diagnostic criteria of the Diagnostic and Statistical Manual of Mental Disorders 4 (DSM-IV) [5]. It consists of nine questions concerning inattentive symptoms and nine questions assessing ADHD-related hyperactivity/impulsivity, as reported in the DSM-IV. Each question is followed by concrete examples, for both childhood and adulthood, to help the patient with identification of specific symptoms. Additionally, the DIVA 2.0 interview investigates the impact of ADHD symptoms on the patient’s functioning in the fields of work/education, relationship and/or family, social contacts, hobbies, and self-confidence/self-image. DSM-IV criteria for ADHD requires at least six symptoms in at least one cluster and requires age of onset before 7 years. The interview usually takes 30 to 90 min. DIVA 2.0 is available in many different languages and free of charge [22]. DIVA 2.0 showed good validity among SUD patients with comorbid ADHD [25]. Furthermore, one study showed that DIVA 2.0 interview has a diagnostic accuracy of 100% when compared with the diagnoses obtained by board-certified psychiatrists [26]. All DIVA 2.0 interviews were conducted by the same interviewer (N.G.).

Recruitment and screening at the VFG site took place between 24 May and 11 June 2021, while recruitment and screening at the DIA site took place between 9 and 23 August 2021. DIVA 2.0 Interviews were held by appointment in the following months. IBM SPSS Version 28.0.1.1 (14)^®^ was used for statistical analysis.

## 3. Results

### 3.1. Sociodemographic and Clinical Characteristics of the Study Participants

The study included 106 participants in total, where 20.8% were female and the mean age was 49.58 ± 7.71 years. In total, 22.6% fulfilled diagnostic criteria for childhood ADHD and 17.9% for persisting adult ADHD using the DIVA 2.0 interview. In 83.3%, ADHD had not been diagnosed before participating in this study.

DIA: 43 out of 54 patients agreed to participate. There was one dropout because one participant did not want to conduct the DIVA interview after completing the questionnaires. In total, 21.4% was female and the mean age was 50.48 ± 7.82 years. Moreover, 45.2% were treated with intravenous diamorphine alone and the remaining individuals additionally received methadone once a day. In total, 16.7% met ADHD diagnosis criteria and in 14.3% ADHD persisted in adulthood. In 85.7% of cases, ADHD had not been diagnosed before participating in this study.OOT: 64 out of 134 patients agreed to participate. There were no dropouts. In total, 20.3% were female and the mean age was 48.98 ± 7.65 years. Among them, 84.4% were treated with l-polamidone, 11% with buprenorphine, 1.6% with methadone, and 3.1% with morphine. Moreover, 26.6% met ADHD diagnosis criteria and in 20.3%, ADHD persisted in adulthood. In 82.4% of cases, ADHD had not been diagnosed before participating in this study.

In summary, almost every fifth study participant suffering from OUD was diagnosed with comorbid adult ADHD, and most of them (83.3%) had been undiagnosed previous to this study.

The two populations show very similar mean age and gender distribution; however, of note, there was a 6% difference in the prevalence of ADHD in adulthood.

### 3.2. WURS-k, ADHD-SR, and DIVA 2.0

Overall, 25.5% met the cut-off criteria for ADHD in WURS-k and 28.3% in ADHD-SR, while 18.9% met the cut-off criteria for ADHD in both. Of 37 DIVA 2.0 interviews, 64.9% confirmed the presence of the ADHD diagnosis, with the combined presentation being the most prevalent.

For DIVA 2.0 subtype results, see Table 1.

In detail:DIA: 19% met the cut-off criteria in WURS-k, 19% in ADHD-SR, and 14.3% met the cut-off criteria in both. Out of 10 DIVA 2.0 interviews, 70% confirmed the presence of ADHD, with the combined presentation being the most prevalent.OOT: 25.5% met cut-off criteria in WURS-k, 29.7% in ADHD-SR, and 20.3% met the cut-off criteria of both inventories. Out of 27 DIVA 2.0 interviews, 63% confirmed the presence of ADHD, with combined presentation being the most prevalent.

With 35.1% of false positive results, WURS-k and ADHD-SR appeared to be a very useful diagnostic tool in this study because of their simple and low-threshold usability. To meet the standards of the international [19] and German [27] guidelines, the ADHD diagnosis must be confirmed as a clinical diagnosis. A structured diagnostic interview, i.e., the DIVA 2.0 Interview, can aid diagnostic assessment and has been very useful in our research setting. In our OUD populations with comorbid ADHD in adulthood, the combined presentation was the most prevalent (63.2%), followed by the predominantly hyperactive-impulsive presentation (21%) and the predominantly inattentive presentation (15.8%).

### 3.3. Urine Samples

We analyzed urine samples of 24 participants. In total, 29.2% of the participants screened negative for all substances. For the prevalence of each substance, see Table 2. Heroin was detected the most and cocaine the least frequently. The OOT clinic usually screens for heroin, benzodiazepines, cocaine, and amphetamines, while the DIA clinic additionally screens for cannabis. Positive cannabis results were found in 71.4% of DIA participants.

These urine analyses show that patients with ADHD tend to abuse substances with sedative properties, such as heroin or benzodiazepines.

## 4. Discussion

In our sample of adult patients with OUD undergoing daily opioid maintenance treatment, we found an ADHD prevalence of 17.9%. This is consistent with previous studies among patients with OUD in other settings [14,28,29] and matches the findings of a meta-analysis published by Katelijne van Emmerik-van Oortmerssen and colleagues in 2012 [13] and the recent meta-analysis published by Thomas Santo and colleagues in 2022 [15]. In our study, in OUD patients receiving opioid maintenance treatment, the prevalence of adult ADHD was lower in those receiving daily intravenous diamorphine maintenance treatment and, in some cases (54.8%), additional methadone treatment, than in those treated with an oral opioid maintenance treatment alone. We postulate various reasons for this novel finding:OUD patients with adult ADHD might benefit less from treatment with intravenous diamorphine, and therefore, do not make use of it.OUD patients with adult ADHD could have difficulties sustaining the intravenous diamorphine maintenance treatment with up to three treatments per day. Thus, they are underrepresented in this special subgroup of OUD patients.

The biggest difference to the ADHD population without OUD is the large proportion of the predominantly hyperactive-impulsive presentation. This difference further increased in the DIA subpopulation. In this subpopulation, the combined presentation was the most prevalent (50%) followed by the predominantly hyperactive-impulsive presentation (33.3%) and the predominantly inattentive presentation (16.7%). Peculiar is the fact that the predominantly hyperactive-impulsive presentation was more prevalent among OUD patients, in particular in the DIA patient cohort, than among patients suffering from ADHD without comorbid OUD.

Further studies directly comparing subtype prevalence in ADHD populations with and without comorbid OUD, as well as studies with more participants in intravenous diamorphine maintenance treatment clinics would be needed to determine if this observation is statistically significant and reproducible. Clinical significance could then be further assessed, e.g., in order to determine whether OUD patients with comorbid ADHD of the hyperactive-impulsive subtype benefit more from diamorphine maintenance treatment than oral opioid maintenance treatment alone.

Our urine analyses suggest that patients with ADHD tend to abuse substances with sedative properties, such as heroin or benzodiazepines. This fits the results of the aforementioned meta-analysis (2012), which demonstrated a higher prevalence of ADHD among patients with OUD and alcohol use disorder than among patients with cocaine use disorder [13]. There were many studies among patients with alcohol use disorder over the years, showing similar results with a high ADHD prevalence (about 20%) [25,29,30,31,32,33]. Recent studies in patients with benzodiazepine use disorder show even higher ADHD prevalence rates, which lie between 31% [34] and 39% [35,36], but there are also studies in populations of patients with cocaine use disorder showing high ADHD prevalence between 14% [37] and 20% [38]. Thus, people with adult ADHD might adjust their substance abuse to their most disruptive ADHD symptoms.

**Limitations:** We examined a small sample cohort of a special subpopulation of OUD outpatients recruited in one region. Future studies should include larger numbers of participants in order to allow for statistical analysis of results as well as recruitment of more diverse populations in different locations.

The higher rate of non-included patients in the OOT than in the DIA populations must be considered in the interpretation.

Diagnosis of ADHD relied on interviewing the patients themselves. An interview with the partner or the parents would have provided even more diagnostic certainty. However, this was not possible within the framework of our study design.

The diagnostic procedure required two steps: first a screening questionnaire and then a structured interview. Thus, underestimation of ADHD prevalence due to false-negative screenings is possible.

Furthermore, additional comorbidities are frequent in OUD and ADHD patients and can complicate diagnostic assessment; the same applies to the frequent abuse of other substances in patients with OUD.

## 5. Conclusions

In our sample, which is as far as we know the first sample ever looking for ADHD in a population of patients with severe OUD on daily diamorphine maintenance treatment, almost every fifth study participant suffering from severe OUD met the diagnostic criteria of adult ADHD, of which 83.3% had been undiagnosed before. Future approaches should implant a routine screening for ADHD among SUD patients and develop new therapeutical concepts for this group of special patients in order to hopefully improve the physical and mental health of patients with ADHD and comorbid SUD. Additionally, early diagnosis of ADHD could attenuate the severity of SUD or even prevent the occurrence of SUD.

In particular, the possible benefit of intravenous diamorphine treatment for patients suffering from severe OUD and comorbid ADHD should be investigated more intensively with larger participant numbers in multicentric study designs.

## Figures and Tables

**Table 1 ijerph-20-02534-t001:** DIVA 2.0 subtype results. Absolute numbers of participants are in parantheses.

Subtype	Total	DIA	OOT
ADHD only in childhood	20.8% (5)	14.3% (1)	23.6% (4)
Combined presentation	50% (12)	42.9% (3)	52.9% (9)
Predominantly inattentive presentation	12.5% (3)	14.3% (1)	11.8% (2)
Predominantly hyperactive-impulsive type	16.7% (4)	28.6% (2)	11.8% (2)

**Table 2 ijerph-20-02534-t002:** Screening results of urine samples from OUD patients with comorbid ADHD. Absolute numbers of participants are in parentheses.

Substance	Total	DIA	OOT
heroin	50% (12)	57.1% (4)	47.1% (8)
benzodiazepines	29.2% (7)	28.6% (2)	29.4% (5)
cocaine	12.5% (3)	14.3% (1)	11.8% (2)
amphetamines	16.7% (4)	14.3% (1)	17.6% (3)

## Data Availability

The data that support the findings of this study are available from the corresponding author upon reasonable request.

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
