# Peer review of "ADHD Prevalence among Outpatients with Severe Opioid Use Disorder on Daily Intravenous Diamorphine and/or Oral Opioid Maintenance Treatment"

_ijerph, 2023, doi:10.3390/ijerph20032534_

Round 1

Reviewer 1 Report

The study covers an interesting and clinically relevant but still under investigated topic. To my knowledge, there is no study on ADHD prevalence in diamorphine treated patients. I very much appreciate the approach of the authors and their pragmatic study design, that can be transferred easily into clinical practice. 

There are a few concerns:

o   DIVA 2.0 is based on DSM-IV criteria for ADHD, but DSM-5 requires less symptoms for adults (at least 5 in at least one cluster), requires less retrospective childhood symptoms and requires onset before age of 12 years (instead of 7). Which criteria were used for ADHD diagnosis?

o   Comorbidity is frequent in ADHD+SUD and can complicate diagnostic assessment, the same applies to the frequent abuse of other substances in individuals with OUD. The pragmatic approach of the study is justified especially in this population, as long as severe withdrawal and intoxication symptoms were excluded (see Crunelle et al., 2018 – international consensus statement on SUD + ADHD). These limitations (that are very hard to overcome in this population) should be shortly mentioned.

o   The diagnostic procedure required two steps: first a screening questionnaire (what were the cut-off scores?) and then a structured interview. Thus, underestimation of ADHD prevalence due to false-negative screenings is possible and should be mentioned in the limitations.

o   What is the benefit of the tables? Probably it would be better to summarize substance across treatment site (e.g. N = X methadone, N = X buprenorphine, N = X diamorphine + methadone, N = X diamorphine …) instead of having small subgroups of different brands. It might be interesting to see screening results (positive) and DIVA results (% ADHD) for different treatment types (substances). See also Lugoboni et al., 2017 https://doi.org/10.1016/j.psychres.2017.01.052

o   I suggest to generally change the approach of the manuscript from reporting results of two different sites separately to reporting results depending on substances used for treatment of OUD (diamorphine, buprenorphine …).

o   I don’t understand table 2: what does childhood ADHD mean – positive DIVA result for childhood ADHD?

o   Table 4 lists drugs found in urin samples – when were these samples acquired? Since the main focus of this manuscript is ADHD prevalence, a comparison between ADHD+ and ADHD- would be more interesting than a comparison between sites.

o   All tables need a proper legend. Always include not only relative (percent) but also absolute numbers.

o   NICE and German guidelines are mentioned in the results section: these references should be introduced in the methods section. Do these guidelines really require structured interviews? ADHD is a clinical diagnosis and other studies (e.g., Luderer et al. 2020 in AUD) did not show 100% validity for the DIVA; however, structured interviews can certainly aid diagnostic assessment and are very useful in research and clinical setting.

o   Also, ADHD subtypes/presentations should be compared with non-OUD populations only in the discussion

o   A 6% difference in ADHD prevalence between the two sites might seem large, but even if it was statistically significant (was it?) the higher rate of non-included patients in the VFG site (could you name it differently?) attenuates the findings.

This should also be mentioned in one sentence in the limitations (more ADHD symptoms = more interest in participating or less ability to show up for diagnostic appointments?).

o   The discussion could focus more on the high rate of previously undetected ADHD. Also, the comparison with other SUDs could be shortened. What would you recommend for clinicians working with patients with OUD - how should they approach diagnostic assessment of ADHD? Also, just one sentence on implications for treatment of ADHD (stimulants?) might be added. 

o   The term opioid maintenance (or agonist) treatment is more common than the term substitution.

o   Semi colons as decimal separators should be replaced by periods throughout the manuscript.

o   The term alcoholics should be replaced by “individuals (or patients) with alcohol use disorder)

 References:

-          Please use the English version of the ICD-10 as reference

-          Are there international publications that could replace ref. 16, 23 and 32?

-          Please add the highly important international consensus statement (Crunelle et al., 2018)

-          A recent meta-analysis showed similar ADHD prevalence rates in OUD https://doi.org/10.1016/j.drugalcdep.2022.109551 which might replace ref. 13

Reviewer 2 Report

Thank you for the opportunity to review this well written report on an important subject. ADHD in patients with opioid use disorder is a subject that has not received enough research attention up to now. I advise some minor changes before publication:

Minor comments:

Introduction:

-          It is not quite clear how diamorphine patients differ from "normal" substitution patients – to my knowledge, diamorphine is mainly used for severely addicted patients for whom the "normal" substitution has not worked.

-          Accordingly, this difference could influence your research questions: Could we have expected higher ADHD prevalence in the diamorphine subsample? Why is it important to assess ADHD prevalence in persons treated with diamorphine?

Results:

-          Table 1: Due to small sample sizes and because the role of substitution substances is not further investigated in this article, I recommend that this table be shortened or deleted altogether

-          Tables 2 und 3: Due to small sample sizes (n = 37 and n = 24), percentages are not the (only) appropriate information. I recommend adding the number of persons in each cell

Spelling:

-          Minor spelling checks: Since I am not a native speaker myself, I can only make cautious suggestions here. But please check the punctuation, and I found some minor typos (e.g., line 30/31: 2of 2,5% in adults and 3.4% in childhood“ (decimal . vs. ,); line 126 “in 14,3 % in persisted in adulthood“ – delete on “in”?)

-          According to APA7, percentages are reported without decimal places
